# Local Adaptive Illumination-Driven Input-Level Fusion for Infrared and Visible Object Detection

**Jiawen Wu** [1,2], **Tao Shen** [1,2,*], **Qingwang Wang** [1,2], **Zhimin Tao** [3,4], **Kai Zeng** [1,2] **and Jian Song** [1,2]

1  Faculty of Information Engineering and Automation, Kunming University of Science and Technology, Kunming 650500, China
2  Yunnan Key Laboratory of Computer Technologies Application, Kunming University of Science and Technology, Kunming 650500, China
3  Beijing Anlu International Technology Co., Ltd., Beijing 100043, China
4  School of Transportation Science and Engineering, Beihang University, Beijing 100191, China
*  Correspondence: shentao@kust.edu.cn; Tel.: +86-0871-65916593

**Abstract:** Remote sensing object detection based on the combination of infrared and visible images can effectively adapt to the around-the-clock and changeable illumination conditions. However, most of the existing infrared and visible object detection networks need two backbone networks to extract the features of two modalities, respectively. Compared with the single modality detection network, this greatly increases the amount of calculation, which limits its real-time processing on the vehicle and unmanned aerial vehicle (UAV) platforms. Therefore, this paper proposes a local adaptive illumination-driven input-level fusion module (LAIIFusion). The previous methods for illumination perception only focus on the global illumination, ignoring the local differences. In this regard, we design a new illumination perception submodule, and newly define the value of illumination. With more accurate area selection and label design, the module can more effectively perceive the scene illumination condition. In addition, aiming at the problem of incomplete alignment between infrared and visible images, a submodule is designed for the rapid estimation of slight shifts. The experimental results show that the single modality detection algorithm based on LAIIFusion can ensure a large improvement in accuracy with a small loss of speed. On the DroneVehicle dataset, our module combined with YOLOv5L could achieve the best performance.

**Keywords:** visible–infrared images; object detection; multimodal fusion; illumination perception; offset estimation

## 1. Introduction

Object detection in remote sensing images is an important step in many applications, such as traffic condition monitoring, road planning, and mountain rescue [1–5]. With the continuous advancement of technology and the continuous expansion of demand, the types of detection scenarios are gradually becoming more complex and diversified. One of the most representative and challenging tasks is to achieve effective detection in scenes with complex illumination conditions. Meanwhile, near-ground remote sensing and unmanned aerial vehicle remote sensing are also developing, but the computing resources of embedded platforms such as vehicles and airborne processors are limited. Ensuring the real-time detection of models mounted on these platforms is also an important issue.

In recent years, the general object detection models have been mainly based on deep learning methods. This includes one-stage methods that directly predict object locations and categories through end-to-end networks, such as SSD series [6,7], YOLO series [8–11], DETR [12], etc., and two-stage methods that first extract the region of interest, and then determine the object location and category, such as Faster R-CNN [13], mask R-CNN [14], DetectoRS [15], etc. These general object detection methods have achieved impressive results on public datasets such as VOC [16] and COCO [17]. The development of object

detection based on UAV remote sensing is driven by the breakthroughs of these algorithms. Zhang et al. pruned YOLOv3 to achieve real-time detection in the ViDrone 2018 dataset [18] and showed good balance accuracy and inference speed [19]. Liu et al. built image pyramids and extracted features from the backbone networks of different depths to improve the small-target recognition ability [20]. Li et al. designed a multi-scale fusion channel attention model based on YOLOv3 to achieve information complementarity between channels [21]. Yu et al. designed the Class-Biased Sampler and Bilateral Box Head components, which use bias against the head class and tail class in order to solve the long-tail distribution problem in UAV images [22]. However, these models take visible images that are obtained during the day or in bright scenes as the training sets. The nighttime and extreme lighting scenes were not considered.

Since infrared imaging reflects the infrared radiation information of the target, it has the advantage of not being limited by the illumination conditions. As shown in Figure 1, under poor illumination conditions, it is difficult for the visible light sensor to capture the scene information, but the infrared sensor can still normally capture the thermal radiation emitted by the targets. Therefore, the reasonable fusion of infrared and visible images helps to improve the around-the-clock detection capability of the model. At present, object detection based on the multimodal fusion of infrared and visible has been widely studied [23–30]. In the task of multimodal object detection, according to the different fusion stages, the multimodal fusions are mainly divided into four categories, i.e, input-level fusion, feature-level fusion, decision-level fusion, and multi-stage fusion.

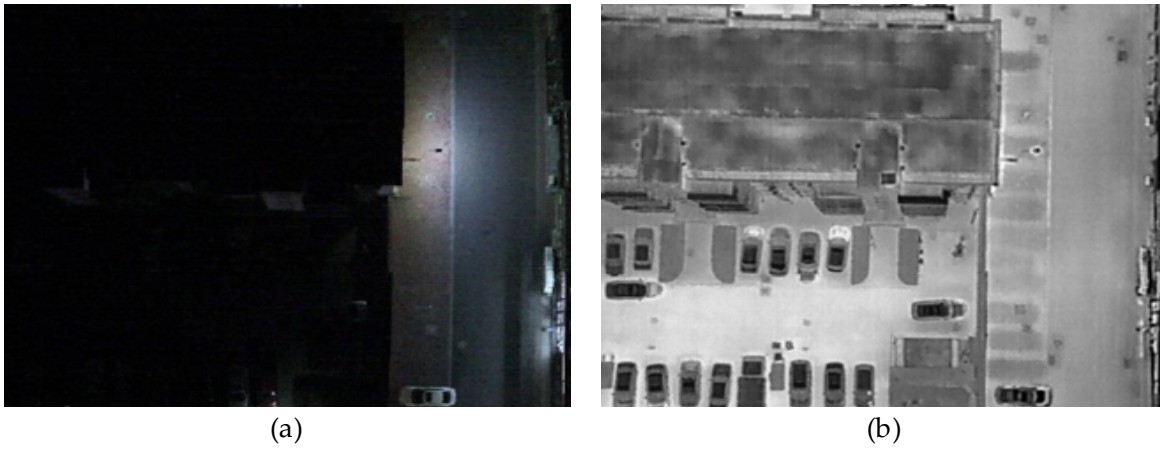

(a)          (b)

**Figure 1.** UAV images are acquired by visible light camera and infrared camera, respectively. (**a**) Visible image. (**b**) Infrared image.

Input-level fusion means that infrared and visible images are firstly fused into an image with two modalities' information, and then the fused image is input into the object detection network for feature extraction and bounding box regression. The advantage of input-level fusion is that it can generate an image that highlights targets and contains more abundant semantic information. To our knowledge, there is currently no object detection method with input-level fusion for remote sensing or assisted driving. Infrared and visible image fusion methods can be regarded as a part of input-level fusion research, but these methods focus on how to obtain better visual quality in images. In particular, image fusion methods based on deep learning have developed rapidly in recent years. Li et al. first designed a fusion model of an encode–decode structure based on deep learning [31]. Zhao et al. decomposed the image into background and details through the encoder, and fused the background and detail information of the two modalities and then input it to the encoder for image reconstruction [32]. Ma et al. first used a GAN network to fuse images, and adaptively learned fusion rules through direct confrontation between the generator and the discriminator in the model [33]. Although these image fusion studies achieved remarkable results, more suitable images for subsequent visual tasks were rarely considered. Moreover,

these image fusion technologies use image pairs under the full alignment assumption. However, in the real scene, the physical characteristics, such as the field of view and parallax between sensors, are different. In the process of data acquisition, there are also problems such as external interference and component aging [29]. These problems cause the inconsistent positions of objects in the same frames of infrared and visible images. The object forms a ghost in the fused image, which affects subsequent detection.

Current research on visible–infrared object detection mainly focuses on feature-level fusion and decision-level fusion. Li et al. designed a dual-input object detection model based on the Faster RCNN framework. They realized multimodal information complementation by jointly using an RPN module and confidence score fusion [25]. Zhang et al. utilized the deep feature stitching results of two modalities to provide attention weights for shallow features [26]. Guan et al. proposed two-stream deep convolutional neural networks and sensed the illumination through FCN to allocate the weight of a two-stream feature map [27]. Zhou et al. proposed a difference module to balance the two modalities of information and a miniature sub-network to perceive the illumination, and then employed the lighting information to weight the confidence scores of the infrared and visible predictions [28]. These methods require two backbone networks to extract features for infrared and visible images separately, which results in more computation and is difficult to deploy in UAV remote sensing. Furthermore, the illumination is usually labeled as day or night in the fusion architecture based on an illumination perception module, and this means that module cannot correctly characterize the illumination of some complex scenes. For instance, when the camera faces the light source, the visible image is overexposed, which leads to partial loss of the light source and its surrounding information. In some specific daytime scenes, the targets on the shaded side are also difficult to distinguish from the background due to insufficient lighting. Furthermore, the current methods often extract the lighting information of the whole image and ignore the differences in lighting in different regions [25,27,28]. This may bring an issue in which the regions that need infrared information cannot be enhanced but some regions are blurred by infrared.

In order to solve the above problems, this paper designs a miniature input-level fusion module for infrared and visible detection. The module achieves the fusion of the multimodal images through limited calculation. Since the perception of lighting information in previous methods is not reasonable, a new labeling method is designed for the estimation of the illumination. Instead of obtaining the lighting information of the whole image, the image is divided into several blocks and the lighting information of each block is perceived. At the same time, considering that multimodal data acquisition is unable to achieve complete alignment, we develop an offset estimation module to assist infrared visible image alignment.

In summary, the main contributions of this paper are as follows:

(1)	A local illumination perception module supervised by pixel information statistics is proposed to fully perceive the illumination difference of each area in the image, which provides a more suitable reference for fusion.

(2)	An offset estimation module composed of bottleneck blocks is designed to predict the location offset of objects in different modalities, which achieves fast alignment between image pairs.

(3)	The proposed local adaptive illumination-driven input-level fusion module (LAIIFusion) can simply convert the single modality object detection model into a multimodal model, and guarantees real-time inference while improving the detection performance.

The rest of this paper is arranged as follows. In Section 2, the overall structure of the model and the specific details of the two proposed modules are described. Section 3 provides the details of the experiments and related analyses. Finally, Section 4 draws specific conclusions and provides some discussion.

## 2. Methods

In this section, we describe the local adaptive illumination-driven input-level fusion approach for infrared and visible object detection. The overall framework of the object detection model and the detection process are first described. Afterwards, the details of the proposed local illumination perception module and offset estimation module are introduced. Finally, the loss function used by the detection framework is introduced.

### 2.1. Overall Network Architecture

Typically, input-level fusion modules consist of an encoder and decoder [31,34]. The purpose of the encoder is to extract the features of two modalities. The decoder is employed to reconstruct and fuse the features. In order to prevent the loss of information caused by downsampling, the size of the feature map is always the same as that of the input image, but this setting consumes more computing resources. In this paper, the target is to obtain an image that can be easily recognized by the detection network, instead of pursuing the visual quality of the image. Therefore, the encoder part is replaced by an efficient information perception method.

The overall framework of the proposed LAIIFusion is shown in Figure 2. The module consists of three submodules: the local illumination perception module, offset estimation module, and adaptive fusion module. In LAIIFusion, the infrared and visible images $I_{inf} \in R^{H \times W \times C}, I_{vis} \in R^{H \times W \times C}$ (H, W, and C, respectively, represent the height, width, and channel) are first downsampled by two convolutional layers with a $3 \times 3$ kernel. Through this operation, the computational complexity of subsequent information perception is reduced while the critical information is preserved. The downsampled infrared and visible images are represented as $I_{ds\_inf} \in R^{h \times w \times C}, I_{ds\_vis} \in R^{h \times w \times C}$, where h = H/4, w = H/4.

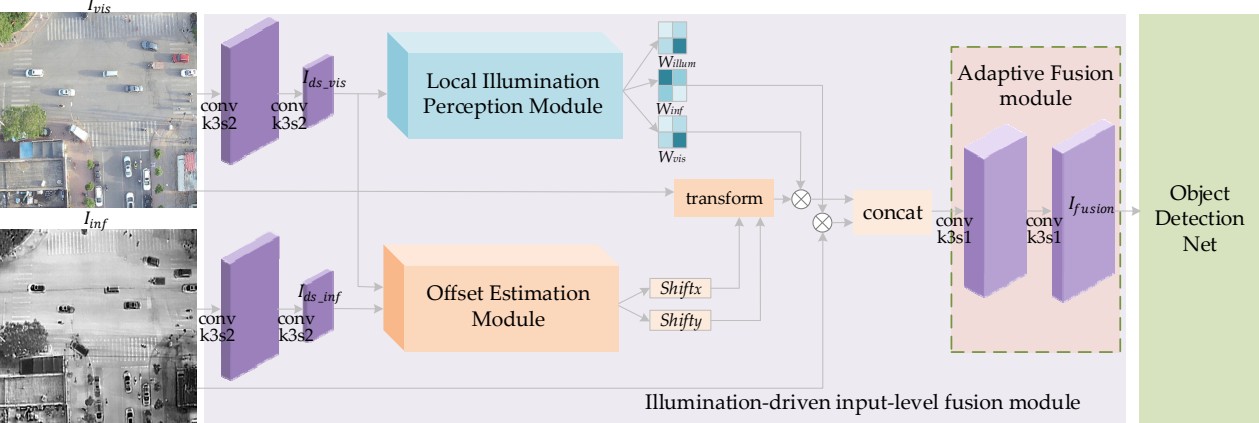

**Figure 2.** The overall framework of the local adaptive illumination-driven input-level fusion module. Conv k3s1 indicates that the convolutional kernel size is 3 and stride is set to 1.

Then, $I_{ds\_vis}$ is input to the local illumination perception module to extract the illumination perception matrix $W_{illum}$ and generate the infrared and visible weight matrix $W_{inf}, W_{vis}$, which is expressed as

$$W_{illum}, W_{inf}, W_{vis} = F_{LIP}(I_{ds\_vis}) \qquad (1)$$

where $F_{LIP}$ denotes the local illumination perception module. This module is trained by calculating the error between the illumination perception matrix and the illumination label. The amount of information input from infrared and visible into the adaptive fusion module is determined by the weight matrix $W_{inf}, W_{vis}$. Moreover, $I_{ds\_inf}, I_{ds\_vis}$ are used to estimate the relative offset $shift_x$, $shift_y$ of the two, which is expressed as

$$shift_x, shift_y = F_{align}(I_{ds\_inf}, I_{ds\_vis}) \qquad (2)$$

where $F_{align}$ represents the offset estimation module. Using $shift_x$, $shift_y$, and affine transformations, $I_{vis}$ is converted into $I'_{vis}$ aligned to $I_{inf}$ [35]. Input $I_{vis\_inf} \in R^{H \times W \times 2C}$ of the adaptive fusion module is represented as

$$I_{vis\_inf} = concat(W_{inf}I_{inf}, W_{vis}I'_{vis})$$
(3)

Finally, in the adaptive fusion module, two convolutional layers with a kernel size of $3 \times 3$ and stride of $1 \times 1$ are set. These two convolutional layers can learn fusion rules that are beneficial to object detection network through training.

### 2.2. Local Illumination Perception Module

The information from infrared and visible images can complement each other since they show different visual characteristics. It is important to determine the information weights of infrared and visible. Some research works use illumination conditions to determine the weights of infrared and visible [25,27,28], but their illumination perception methods ignore the complexity of the scene. Therefore, we designed a local illumination perception module with more accurate area and illumination labels.

#### 2.2.1. Local Illumination Perception

The illumination conditions of each area in the image are different. To accurately perceive the illumination of each area in the image, the image is divided into grid cells. Therefore, the image I could be expressed as

$$I = \begin{pmatrix} A_{11} & \cdots & A_{1n} \\ \vdots & A_{ij} & \vdots \\ A_{n1} & \cdots & A_{nn} \end{pmatrix}$$
(4)

where $A_{11}, \ldots A_{ij}, \ldots, A_{nn}$ represent the areas of the image. It is equally divided into n parts in the horizontal and vertical directions. Then, the illumination perception matrix $W_{illum}$ is expressed by

$$W_{illum} = \begin{pmatrix} w_{11} & \cdots & w_{1n} \\ \vdots & w_{ij} & \vdots \\ w_{n1} & \cdots & w_{nn} \end{pmatrix}$$
(5)

where $w_{11}, \ldots w_{ij}, \ldots, w_{nn}$ correspond to the illumination perception values of areas $A_{11}, \ldots A_{ij}, \ldots, A_{nn}$, respectively.

Details of the local illumination perception module are shown in Figure 3. Differing from object detection, illumination perception prefers color intensity to texture, semantics, and scale. Since visible images effectively reflect the color intensity of each pixel, downsampled visible image $I_{ds\_vis}$ is utilized as the input of the illumination perception. The input three-channel image is converted into a high-dimensional feature map through convolution. Then, this feature map enters two branches: the perception branch and adjustment branch. The perception part consists of four convolutional layers, one maxpooling layer, and one adaptive average pooling layer. After extracting features, the size of the input feature map is compressed to $16 \times 16$ by adaptive averaging pooling. Next, the illumination perception values for each region are obtained by a convolutional layer with a kernel size of $4 \times 4$ and stride of $4 \times 4$. The adjustment branch is similar to the perception branch except that the last two layers pass through the sigmoid activation function. The adjustment branch outputs an independent prediction matrix $W_0$ for the current scene. Under good illumination conditions, appropriate infrared information is still useful for judgment. Using light perception alone to determine the weights of two modalities is too extreme. Therefore,

at the end of the module, we construct an adjustment function to generate infrared and visible weights. The adjustment function is as follows:

$$\hat{W}_d = \left(\frac{W_d - W_n}{2}\right) \cdot (\gamma \cdot W_0 + \beta) + \frac{1}{2} \tag{6}$$

where $W_d$ and $W_n$ represent the tendency of the illumination branch to judge the scene in the daytime and at night, respectively, and $\gamma$, $\beta$ are two learning parameters initialized with 0 and 1. $W_{vis} = \hat{W}_d$ and $W_{inf} = 1 - \hat{W}_d$ are set as the weights for visible and infrared. The $W_0$ in the adjustment function means that the illumination perception value and visible weight no longer have a simple linear relationship, and it prevents the weight imbalance of infrared and visible under extreme illumination conditions. Moreover, it is obtained from multi-layer convolution prediction, which can more accurately adjust the visible weights of different scenes than the learnable parameters. The images of two modalities are input into the adaptive fusion module based on this weight.

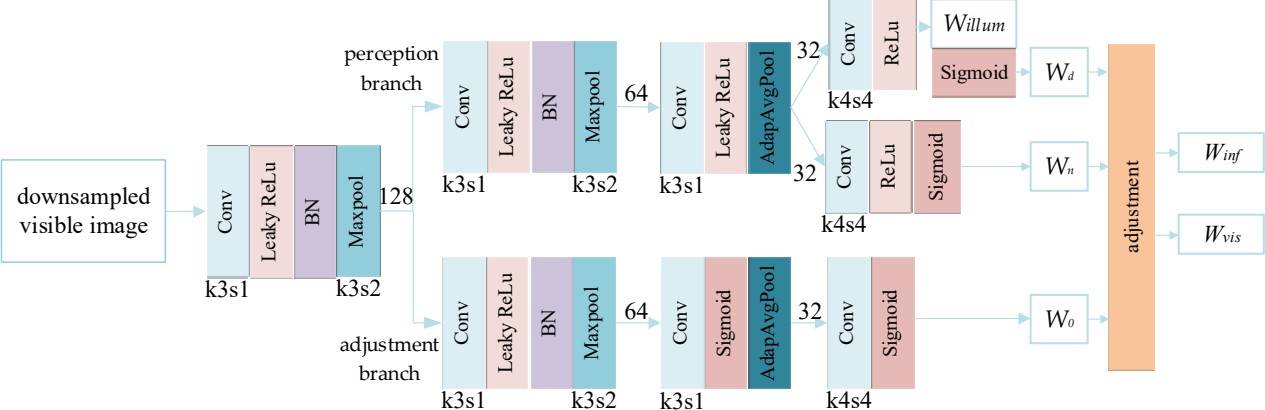

**Figure 3.** Interior structure of local illumination perception module. k3s1 indicates that the kernel size is 3 and stride is set to 1. The number on the connecting line represents the number of channels of the output feature map.

### 2.2.2. Illumination Label Design

The existing methods regard illumination perception as a binary classification problem, distinguishing illumination conditions as day and night. However, the binary classification cannot accurately describe the complex illumination conditions in the real scene. In this paper, the illumination is defined as a specific value, which is calculated from the RGB values of all pixels in an area. The local illumination perception module is supervised by the error between the prediction and the design value of each area. Illumination perception is regarded as a regression problem to eliminate this incompleteness.

During label design, the image is divided into grid cells by default. Specifically, RGB brightness values from 0 to 255 are divided into N intervals. Then, the RGB values for all pixel points in the current area are counted. These statistics in the same interval are added. Next, we find the interval with the highest frequency, and if the interval is the k-th interval, the illumination value of this area is expressed as $w_{label} = k/N$. Figure 4 shows the results of the division, the corresponding illumination values for each area, and the histogram statistics for one area. The illumination error is defined as

$$L_i = \frac{1}{N^2} \| W_{illum} - W_{label} \|_2 \tag{7}$$

where $W_{label}$ represents the label matrix corresponding to the entire image. $W_{illum}$ represents the illumination perception matrix.

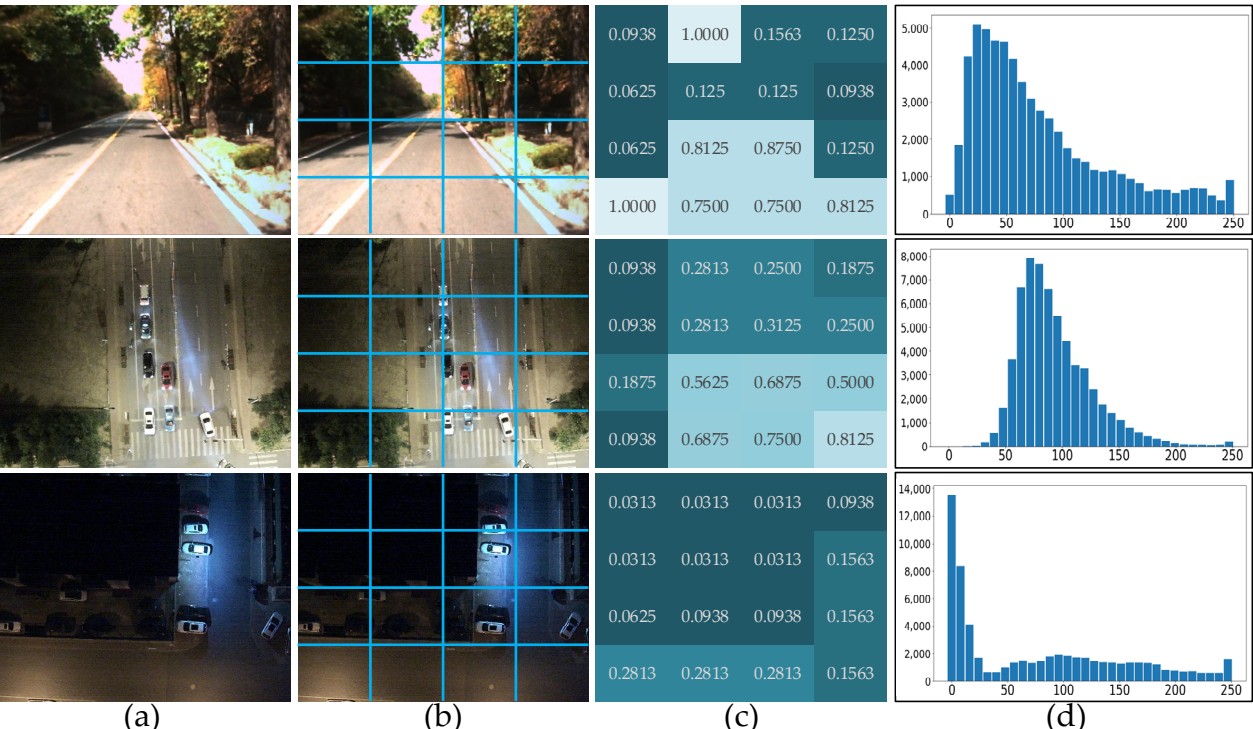

**Figure 4.** (**a**) Original image. (**b**) The original image is divided into grid cells, and the sample image is divided into 16 grid cells. (**c**) The design value of the illumination condition in each area of the image is shown, where the value 1 represents the strongest and the value 0 represents the weakest. (**d**) The histogram of the pixels in the third block of the second row of the partition, where the RGB values are divided into 32 intervals for statistics.

### 2.3. Offset Estimation Module

To capture the spatial offset caused by the process of data acquisition, we design an end-to-end offset estimation module. Figure 5 describes the architecture of the offset estimation module. First, $I_{ds\_inf}$ and $I_{ds\_vis}$ are spliced on the channel dimension. Afterwards, the infrared and visible feature point sets $X_{inf}, X_{vis}$ could be obtained. The collection of these two feature point sets can be represented as

$$X_{inf} \oplus X_{vis} = \text{flatten}(\Phi(I_{ds\_inf} \oplus I_{ds\_vis})) \tag{8}$$

where $\Phi$ is the feature extraction network. We consider image pair $I_{ds\_inf}, I_{ds\_vis}$ as inputs to the network. The output feature map is expanded by a flatten operation to form the feature point set X. Symbol represents channel dimension stitching. The bottleneck structure [36] shown in Figure 6 is utilized to extract features. More critical features can be extracted by mapping the input to high-dimensional space. The bottleneck structure can reduce the computational complexity, and the skip connection can prevent the gradient disappearance. Finally, the shifts $s_x, s_y$ are estimated by two neurons in the linear layer.

The resulting offset is the relative spatial offset between the visible and the infrared images. Therefore, the affine matrix $\theta$ is expressed as

$$\theta = \begin{pmatrix} 1 & 0 & s_x \\ 0 & 1 & s_y \end{pmatrix} \tag{9}$$

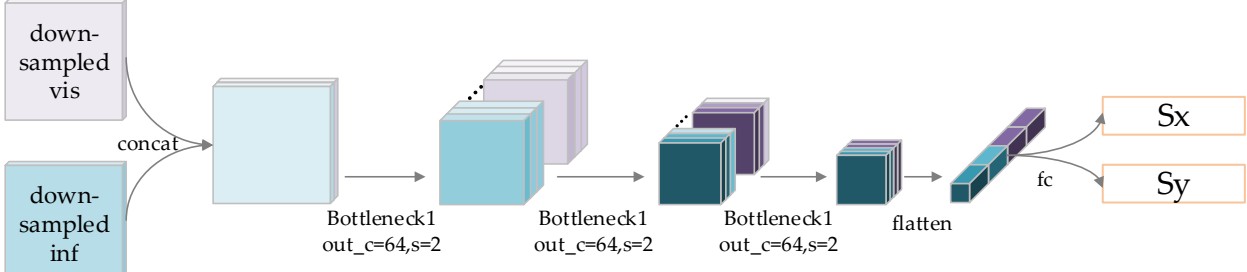

**Figure 5.** Process of offset estimation module.

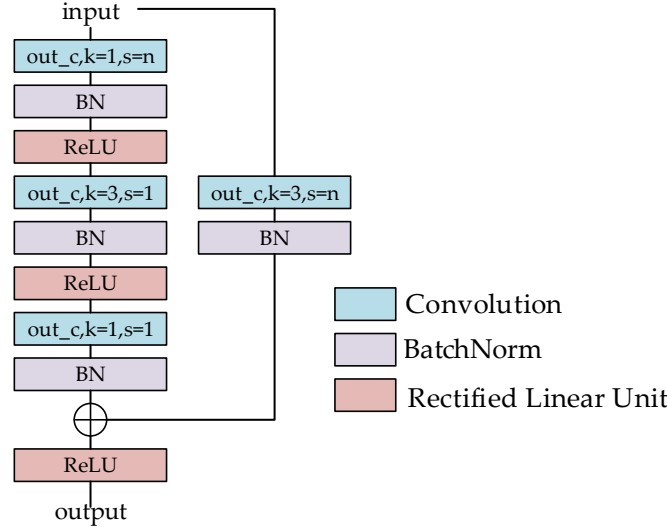

**Figure 6.** Bottleneck structure. In a convolutional block, the first parameter is the output channel, the second parameter is the convolutional kernel size, and the third parameter represents the step size.

This matrix means that the image is shifted $s_x$ in the X axis and $s_y$ in the Y axis. Then, each pixel offset matrix is generated from the affine matrix. For the coordinates x, y of each pixel, the new coordinates $x'$, $y'$ are formed after the following transformation:

$$\begin{pmatrix} x' \\ y' \end{pmatrix} = \begin{pmatrix} 1 & 0 & s_x \\ 0 & 1 & s_y \end{pmatrix} \begin{pmatrix} x \\ y \\ 1 \end{pmatrix} \tag{10}$$

For pixel values of non-integer coordinate points, bilinear interpolation is required to extract pixel values. The calculation formula of the gray value calculated by bilinear interpolation is as follows:

$$V_i^c = \sum_n^H \sum_m^W U_{nm}^c \max\left(0, 1 - \left|x' - m\right|\right) \max\left(0, 1 - \left|y' - n\right|\right) \tag{11}$$

where $V_i^c$ is the gray value of a point on the c-th channel of the output image, and $U_{nm}^c$ is the gray value of the c-th channel point (n, m) of the input image. This formula means that the gray value of the target point $(x',y')$ is determined by the gray value of the four points around (x, y), and the weight is affected by the distance between the two points. Finally, the aligned image $I'_{vis}$ can be obtained by transforming the input image according to the pixel offset matrix and the pixel values of the corresponding coordinate points.

*2.4. Loss Function*

The proposed method is a fusion module for object detection tasks. The training of LAIIFusion is different to that of the other input-level fusion methods. It is trained jointly with the object detection network and is affected by the loss function of the network. The adaptive fusion module in LAIIFusion is guided by detection loss and illumination factors to generate the appropriate image for detection. Therefore, the training loss function of the whole visible–infrared object detection framework consists of at least three parts: illumination perception loss $L_i$, classification loss $L_{cls}$, and regression loss $L_{reg}$. The final loss function is

$$L = L_i + L_{cls} + L_{reg} \tag{12}$$

The $L_{cls}$ and $L_{reg}$ settings are provided by the selected object detection network.

### 3. Experiment and Analysis

*3.1. Dataset Introduction*

The DroneVehicle dataset [37] contains 28,439 pairs of visible–infrared images and all the images are captured by an UAV. This is the first multispectral large-scale vehicle detection dataset from an UAV perspective. Besides day and night scenes, this dataset also includes dark night. Dark night data mainly come from parking lots, residential areas, and road scenes without street lights. Night data come mainly from streets, neighborhoods, and so on. For this dataset, the affine transformation and region clipping methods have been employed to ensure that most image pairs are aligned. However, there are still some image pairs with shifts due to imperfections of the registration algorithm. Figure 7 shows the visualization of images in the DroneVehicle dataset.

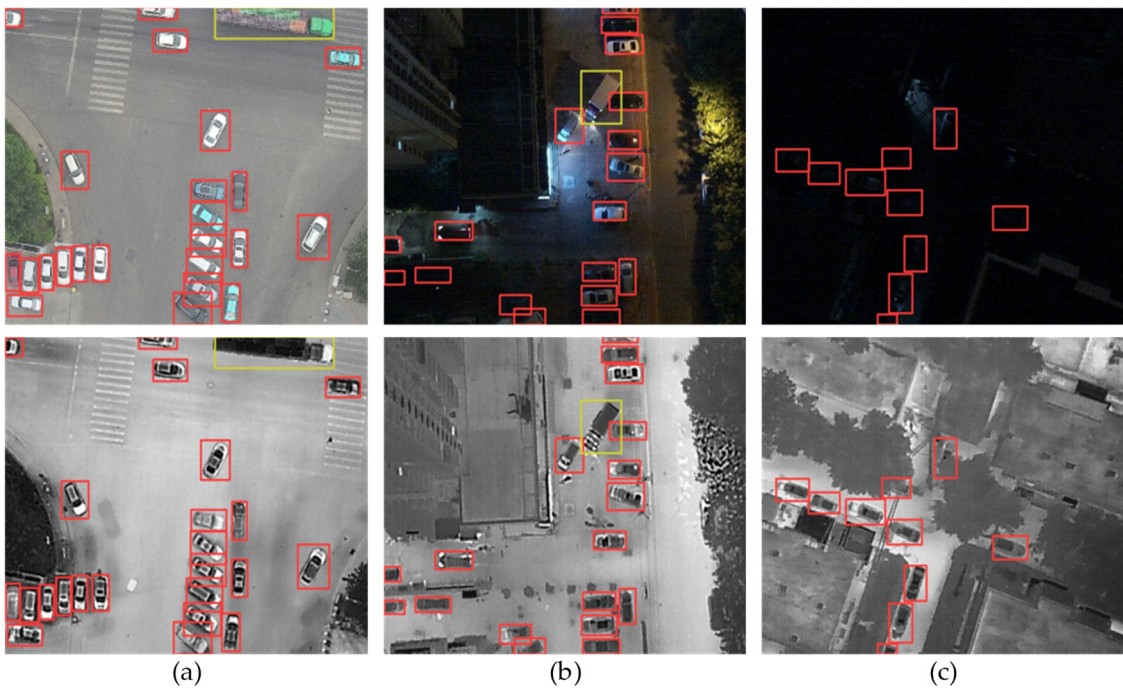

(a)  (b)  (c)

**Figure 7.** Visualization of images in the DroneVehicle dataset. (**a**) Day scene. (**b**) Night scene. (**c**) Dark night scene.

The KAIST multispectral pedestrian dataset [38] contains 95,328 pairs of visible–infrared images, which are captured from a vehicle's top view. The detected objects are classified into three categories: person, people, and cyclist. Different from other datasets, the KAIST dataset not only incorporates a large number of night-captured images to increase the richness of the scenes in the dataset, but also considers the complexity of the illumination conditions in the real scene. Since most of the original KAIST dataset images

are adjacent frames and have high similarity, some images are removed, and the optimized dataset consists of 8963 pairs of training images and 2252 pairs of test images. Some images of the KAIST dataset are shown in Figure 8.

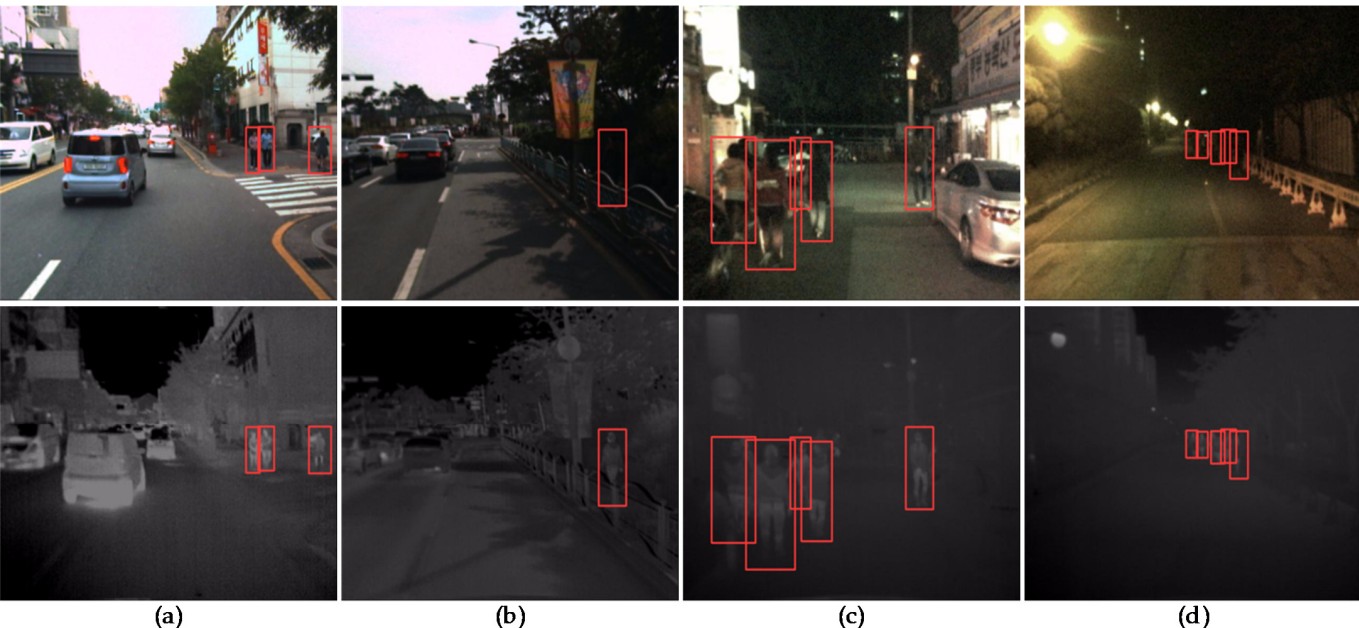

**Figure 8.** Visualization of images in the KAIST dataset. (**a**) Bright daytime scene. (**b**) Daytime scene with shadows. (**c**) Bright night scene. (**d**) Night scene.

### 3.2. Implementation Details

The experiments are conducted in the PyTorch 1.8.0, CUDA 10.2 framework based on a single NVIDIA Tesla V100 and the Ubuntu 20.04 system. The proposed module is combined with YOLOv5L [10] for joint training and performance demonstration. YOLOv5L utilizes the COCO pre-training weight. The model was trained with the SGD optimizer, and 16 batch image pairs were input at a time. The displayed results are the best in the training stage. Linear preheating and cosine annealing are used to update the learning rate during training. When using the KAIST dataset for training, the initial learning rate is 0.0032, and the momentum is 0.843, totaling 20 epochs. On the DroneVehicle dataset, the initial learning rate is 0.01, the momentum is 0.937, and a total of 50 epochs were trained. In addition, all data augmentation methods used in YOLOv5L are cancelled because the method in this paper requires processing of the original image. In contrast experiments, the related methods do not use data augmentation. The IoU threshold for MNS is set to 0.425 during the test.

### 3.3. Evaluation Metrics

For the DroneVehicle dataset, AP is used as the evaluation index. By calculating the area enclosed by the precision (P)–recall (R) curve and the x and y axes, the average precision can better represent the network's detection performance. These indicators are defined as follows:

$$P = \frac{TP}{TP + FP} \tag{13}$$

$$R = \frac{TP}{TP + FN} \tag{14}$$

$$AP = \int_0^1 P(R)dR \tag{15}$$

where FP indicates the number of negative samples misjudged as positive, TP indicates the number of positive samples correctly judged, FN indicates the number of positive samples

misjudged as negative, and TN indicates the number of negative samples correctly judged. For AP, this paper uses an IoU threshold of 0.5, expressed as $AP_{50}$.

In the KAIST dataset, $MR^{-2}$ is utilized to evaluate the effectiveness of network detection. $MR^{-2}$ is an important indicator in pedestrian detection, which consists of the false positives per image (FPPI) and miss rate (MR). The false positives per image (FPPI) represents the average false detection rate for each picture. The miss rate (MR) indicates the miss rate in the test results. Their formulas are expressed as

$$MR = 1 - R \tag{16}$$

$$FPPI = \frac{FP}{the\,number\,of\,image} \tag{17}$$

$MR^{-2}$ is computed by averaging the MR at nine FPPI rates evenly spaced in log-space in the range $10^{-2}$ to $10^{0}$ (for curves that end before reaching a given FPPI rate, the minimum miss rate achieved is used). The smaller the $MR^{-2}$, the better the model performance. In pedestrian detection, $MR^{-2}$ is selected instead of AP because the upper limit of the acceptable false negative rate for each image is independent of the pedestrian density [39].

In addition, the frames per second (FPS) is introduced to measure the computational complexity of the network. FPS is defined as the number of pictures processed per second, and a higher number on the same platform means lower computational complexity of the network.

### 3.4. Analysis of Results

In this section, our experiments are described from three perspectives. First, we compare LAIIFusion with other existing works on the DroneVehicle and KAIST datasets. Second, the effect of different RGB interval division N on the local illumination perception module is analyzed. Then, LAIIFusion is combined with some general object detection models to demonstrate the generality of the proposed method. Finally, the effectiveness of local illumination perception, offset estimation, and LAIIFusion is verified through an ablation study. In these experiments, the best results are shown in bold.

#### 3.4.1. Experiments on DroneVehicle

The proposed LAIIFusion is compared with CMDet and UA-CMDet, which are DroneVehicle dataset benchmarks [37]. We also use an advanced feature-level fusion method, MBNet [28]. It is also compared with an input-level fusion method, addition. The addition method means that infrared and visible images are element-wise added with a weight of 0.5. The size of the input image for all networks is 640 px × 512 px.

Table 1 shows the results of the evaluation on the DroneVehicle dataset. The results show that the proposed LAIIFuison achieves the highest mAP of 66.32%, which is 2.22% better than the benchmark UA-CMDet. Table 2 shows the quantitative results of each category on the DroneVehicle dataset. The proposed method achieves the best detection performance for targets such as buses and freight cars. Figure 9 illustrates the visual comparison with only infrared images, only visible images, the addition method, and the proposed LAIIFusion detection results on the DroneVehicle dataset. It can be observed that the objects are successfully detected by LAIIFusion. Our method can provide more detailed visible information in the area with sufficient light, and generates more recognizable features for the target. Especially in the two images taken at night, buildings and road signs are wrongly identified as vehicles by other methods, and the proposed method can avoid this problem.

**Table 1.** Comparisons of detection performance on the DroneVehicle dataset.

| Fusion Stage | Method | Modality | mAP |
|---|---|---|---|
| Multi-Stage Fusion | CMDet [37] | visible + infrared | 62.58 |
| | UA-CMDet [37] | visible + infrared | 64.01 |
| | MBNet [28] | visible + infrared | 62.83 |
| No Fusion | YOLOv5L [10] | visible | 57.02 |
| | YOLOv5L | infrared | 62.93 |
| Input-Level Fusion | YOLOv5L + Addition | visible + infrared | 64.89 |
| | YOLOv5L + LAIIFusion (ours) | visible + infrared | **66.23** |

**Table 2.** Per-class comparisons of detection performance on the DroneVehicle dataset.

| Method | Car | Truck | Bus | Van | Freight Car |
|---|---|---|---|---|---|
| UA-CMDet [37] | 87.51 | **60.70** | 87.08 | **37.95** | 19.04 |
| YOLOv5L (visible) [10] | 86.83 | 55.50 | 83.14 | 34.42 | 25.34 |
| YOLOv5L (Infrared) | **95.26** | 51.03 | 89.39 | 27.15 | 51.83 |
| YOLOv5L + Addition | 95.19 | 53.66 | 88.56 | 29.74 | 57.29 |
| YOLOv5L + LAIIFusion (ours) | 94.45 | 54.38 | **90.46** | 33.89 | **57.91** |

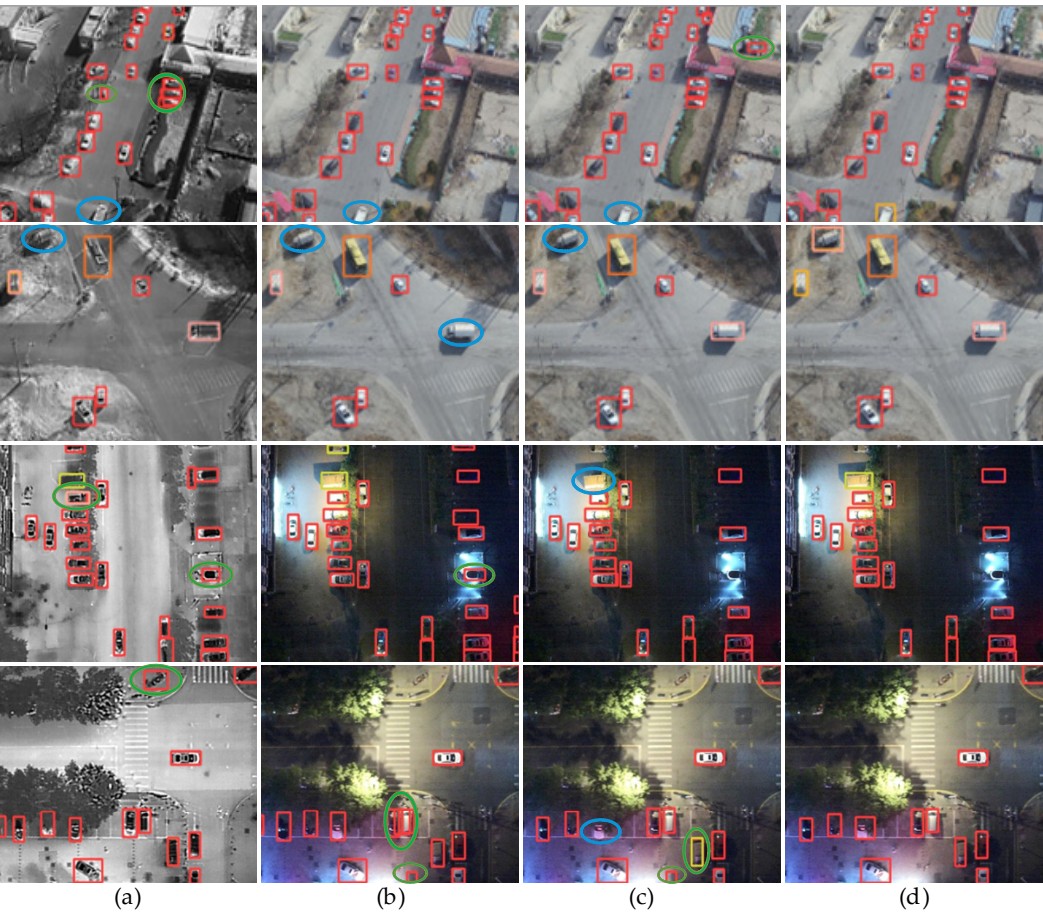

(a)　　　　　　　　(b)　　　　　　　　(c)　　　　　　　　(d)

**Figure 9.** Visualization of only infrared images (**a**), only visible images (**b**), addition method (**c**), and proposed LAIIFusion (**d**) detection results on DroneVehicle dataset. The missed object is marked with blue circles. The incorrect object is marked with green circles. Red, yellow, pink and orange rectangles represent cars, vans, freight cars and bus targets, respectively.

3.4.2. Experiments on the KAIST Dataset

The KAIST dataset is used to verify the effectiveness of the proposed LAIIFusion in more scenarios and applications. For performance comparison, some existing feature-level and decision-level fusion detectors, such as IAF R-CNN [25], CIAN [26], AR-CNN [29], and MBNet [28], are employed to perform detection experiments. It is also compared with two input-level fusion methods, addition and DenseFuse [31]. The performance of these input-level fusion methods is analyzed with the conditions of taking the fused images as YOLOv5L inputs. The size of the input image for all networks is 640 px × 512 px.

Table 3 shows the validation results on the KAIST dataset, where the platform column represents the computing platform used for the network test. The results show that the performance of the proposed method is significantly improved, especially in the night recognition. The night $MR^{-2}$ of the proposed method reaches 6.96, which is 0.9 lower than the second-best detector, MBNet. Although there is still a gap between feature-level fusion and our input-level fusion method in detection performance, the inference speed based on YOLOv5L is 21 FPS faster than MBNet. Moreover, the proposed LAIIFusion realizes real-time detection on the NVIDIA RTX2060 platform. This shows that input-level fusion has lower computational complexity than feature-level fusion, and our method is lightweight. Compared with other input-level fusion methods, LAIIFusion brings the most significant improvement in around-the-clock detection ability, and has the least impact on the inference efficiency of the original model.

**Table 3.** Comparisons of detection performance on the KAIST dataset. In the 'Platform' column, NVIDIA GTX 1080 Ti has 11.3 T single-precision floating point operations (FLOPs), while NVIDIAI RTX 2060 has 6.45 TFLOPs. YOLOv5L (visible) means that only visible images are used for training and testing.

| Fusion Stage | Method | All | Day | Night | Near | Medium | Far | Platform | fps |
|---|---|---|---|---|---|---|---|---|---|
| Multi-Stage Fusion | IAF R-CNN [25] | 15.73 | 14.55 | 18.26 | 0.96 | 25.54 | 77.84 | - | - |
| | CIAN [26] | 14.12 | 14.77 | 11.13 | 3.71 | 19.04 | 55.82 | GTX 1080 Ti | 14 |
| | AR-CNN [29] | 9.34 | 9.94 | 8.38 | 0 | 16.08 | 69.00 | GTX 1080 Ti | 8 |
| | MBNet [28] | **8.13** | **8.28** | 7.86 | **0** | **16.07** | 55.99 | RTX 2060 | 10 |
| No Fusion | YOLOv5L (visible) [10] | 15.14 | 11.24 | 22.97 | 1.82 | 23.05 | 54.70 | RTX 2060 | **32** |
| | YOLOv5L (infrared) | 15.94 | 20.52 | 7.02 | 3.11 | 19.49 | 37.34 | RTX 2060 | 32 |
| Input-Level Fusion | YOLOv5L + Addition | 13.72 | 11.90 | 17.13 | 0 | 22.58 | 63.36 | RTX 2060 | 32 |
| | YOLOv5L + DenseFuse [31] | 16.09 | 14.41 | 19.45 | 0 | 24.42 | 64.36 | RTX 2060 | 18 |
| | YOLOv5L + LAIIFusion (ours) | 10.44 | 12.22 | **6.96** | 1.66 | 16.09 | **40.95** | RTX 2060 | 31 |

In order to show the effectiveness of the proposed LAIIFusion, Figure 10 visualizes only infrared images, only visible images, the addition method, and the detection results of LAIIFusion on the KAIST dataset. It can be seen that our method has better detection accuracy and robustness. Although the addition method uses infrared and visible information, the supplemented amount of infrared information in areas with poor illumination conditions is still insufficient, resulting in missed detection and false detection.

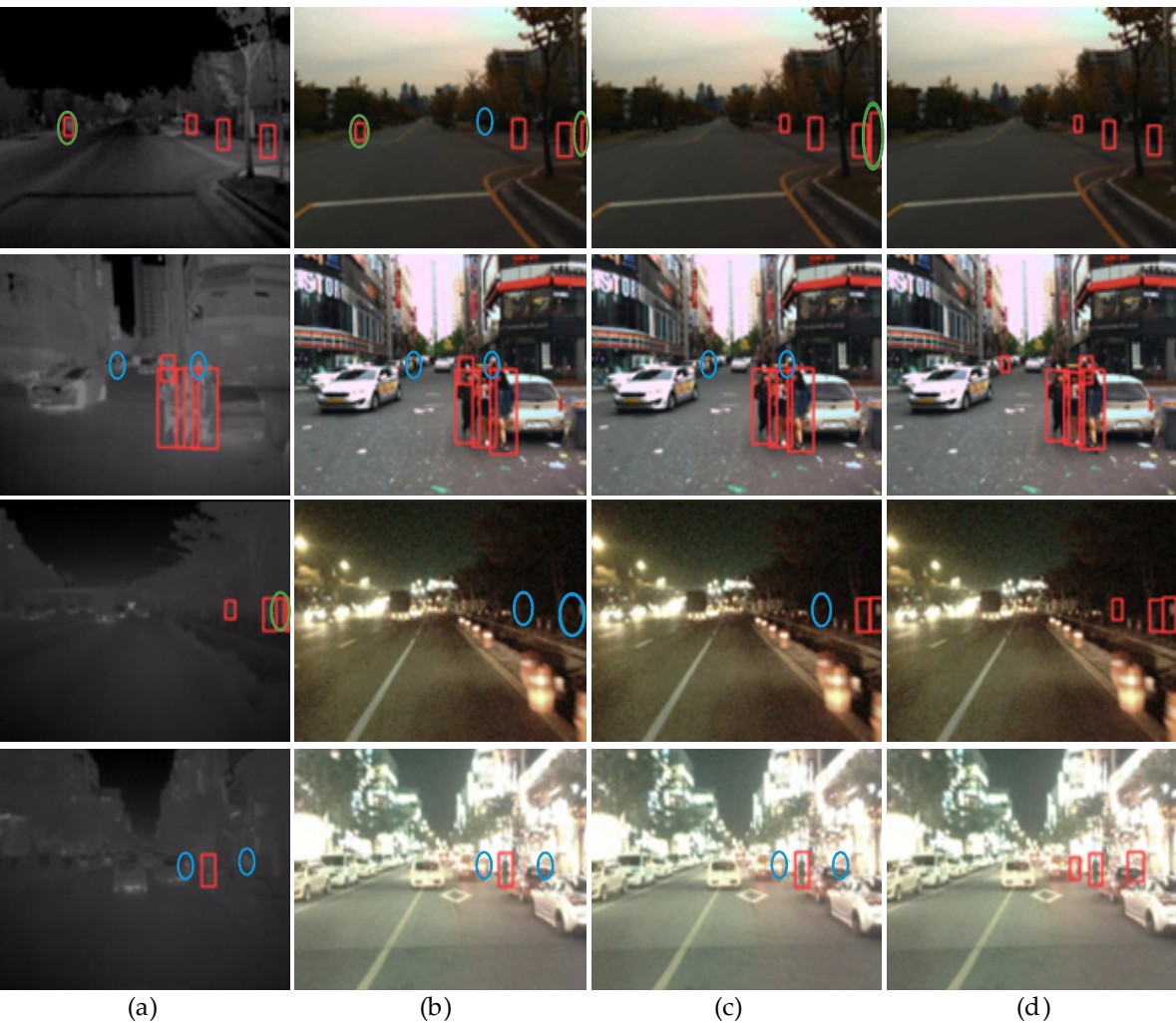

(a)    (b)    (c)    (d)

**Figure 10.** Visualization of only infrared images (**a**), only visible images (**b**), addition method (**c**), and proposed LAIIFusion (**d**) detection results on KAIST dataset. The missed object is marked with blue circles. The incorrect object is marked with green circles. Red rectangles represent the detected pedestrian target.

3.4.3. Parameter Analysis of RGB Value Interval Division

Selecting the number of RGB value interval divisions N is an important step in illumination label design, and it is also the key to ensuring the accurate training of the illumination perception (LIP) module. In order to find the most suitable N, the LIP module is combined with YOLOv5L for joint training and performance demonstration on the DroneVehicle dataset. Table 4 shows that the best performance can be obtained when the RGB value is divided into 32 intervals. More intervals lead to overfitting of training and decrease the robustness of the LIP module. When fewer intervals are employed, the performance begins to decline because the generated illumination label cannot give accurate supervision.

**Table 4.** Result of varying the number of RGB value interval divisions.

| N | mAP | mAP$_{0.5:0.95}$ |
|---|---|---|
| 8 | 64.74 | 46.83 |
| 16 | 64.55 | 46.55 |
| 32 | **65.05** | **48.11** |
| 64 | 64.75 | 47.44 |
| 128 | 64.60 | 47.47 |

### 3.4.4. Generality of Proposed Method on KAIST

To illustrate the generality of our fusion method, LAIIFusion is integrated into recent single modality detectors, including YOLOv3 [9], SSD [6], and YOLOXm [11]. Table 5 demonstrates the detection results of different object detectors with our LAIIFusion. The results show that LAIIFusion has a positive effect on these single-mode detectors and is more effective than adding the two-mode data alone. Compared with the results of the visible modality, LAIIFusion reduces the $MR^{-2}$ of YOLOv3, SSD, and YOLOXm by 8.8, 4.82, and 3.74, respectively. In particular, LAIIFusion can greatly improve the night detection performance of the network. The night $MR^{-2}$ of YOLOv3 decreases from 29.40 to 7.77, and the night $MR^{-2}$ of SSD decreases from 52.90 to 21.75, with a decrease of more than 50%.

**Table 5.** Results of visible modality, addition, and LAIIFusion on KAIST dataset.

| Method | All | Day | Night | Near | Medium | Far |
|---|---|---|---|---|---|---|
| YOLOv3 (visible) [9] | 21.77 | 17.09 | 29.40 | 9.32 | 26.36 | 56.59 |
| YOLOv3 + addition | 18.89 | 21.85 | 13.21 | 0 | 26.92 | 62.55 |
| YOLOv3 + LAIIFusion | **12.97** | **15.50** | **7.77** | **0** | **19.66** | **47.23** |
| SSD (visible) [6] | 37.77 | **30.74** | 52.90 | 11.82 | 41.34 | 77.32 |
| SSD + addition | 35.51 | 31.10 | 44.15 | **4.74** | 46.53 | 82.80 |
| SSD + LAIIFusion | **32.95** | 37.86 | **21.75** | 8.51 | **38.99** | **73.58** |
| YOLOXm (visible) [11] | 21.31 | **14.88** | 33.97 | 1.55 | 28.72 | 64.46 |
| YOLOXm + addition | 22.95 | 20.96 | 27.01 | **0.03** | 34.82 | 72.72 |
| YOLOXm + LAIIFusion | **17.57** | 21.91 | **9.12** | 1.97 | **26.67** | **58.46** |

### 3.4.5. Ablation Study

In this part, we conducted experiments to evaluate the effectiveness of our two submodules on the DroneVehicle dataset and KAIST dataset. First, the effect of YOLOv5 under only visible and only infrared modalities are tested. Table 6 shows that the night detection performance under the visible modality and daytime performance under the infrared modality are both poor. Then, after adding the adaptive fusion (AF) module, the around-the-clock detection performance of the network is significantly improved, and the $MR^{-2}$ drops to 12.23. The AF module provides more balanced detection performance, and the daytime $MR^{-2}$ is 6.11 lower than for the infrared modality. The performance of the model is further improved by adding the local illumination perception (LIP) module, and the around-the-clock $MR^{-2}$ drops to 11.75. However, misalignment leads to noise in the fused image, which results in a slight decline in daytime detection performance. After adding the offset estimation (OE) module, a complete LAIIFusion is formed. With the addition of LAIIFusion, the around-the-clock detection performance of the network further decreased from 11.75 $MR^{-2}$ to 10.44 $MR^{-2}$. Both day and night detection performance are improved. LAIIFusion in the DroneVehicle dataset is 9.33% higher than only the visible modality and 3.42% higher than only the infrared modality. The experimental results show that these modules have good compatibility and generality, and can effectively improve the performance of the entire network.

**Table 6.** Ablation results of adaptive fusion (AF), the local illumination perception (LIP), and offset estimation (OE) module.

| Method | DroneVehicle Dataset | KAIST Dataset | | | | | |
|---|---|---|---|---|---|---|---|
| | mAP | All | Day | Night | Near | Medium | Far |
| YOLOv5L [10] (visible) | 57.02 | 15.14 | 11.24 | 22.97 | 1.82 | 23.05 | 54.70 |
| YOLOv5L (infrared) | 62.93 | 15.94 | 20.52 | 7.02 | 3.11 | 19.49 | **37.34** |
| YOLOv5L + AF | 64.72 | 12.23 | 14.41 | 7.75 | 0.03 | 19.48 | 53.88 |
| YOLOv5L + AF + LIP | 63.71 | 11.75 | 14.14 | 7.33 | **0.02** | 17.51 | 50.66 |
| YOLOv5L + AF + LIP + OE | **66.23** | **10.44** | **12.22** | **6.96** | 1.66 | **16.09** | 40.95 |

## 4. Discussion

### 4.1. Lightweight Technology for Infrared and Visible Object Detection

The object detection model based on infrared and visible has been studied by many scholars because of its advantages, such as not being limited by the illumination conditions, good robustness, etc. However, the demand for a large number of computing resources is the limitation of multimodal models, and there is little research on a multimodal model with a low computational cost. At present, the lightweight methods for deep neural networks mainly include compressing existing models and redesigning lightweight networks. In this paper, the LAIIFusion module is designed to reduce the cost of multimodal feature extraction. The experimental results show that the proposed method achieves real-time multimodal object detection on the NVIDIA RTX 2060 platform. However, the computation of the original network is still large, and it is difficult to achieve real-time inference on small embedded devices such as FPGA and PLC. Therefore, it is still necessary to appropriately use pruning [40,41], quantification [42,43], knowledge distillation [44,45], and other model compression methods to reduce the amount of parameters and calculations of the current model.

### 4.2. Adaptability in Different Around-the-Clock Scenarios

The around-the-clock datasets of two different application scenarios were used to verify the adaptability of LAIIFusion. It can be seen that our method offers the single-mode target detection network more competitive detection performance. Especially when YOLOv5L is combined with LAIIFusion, the around-the-clock vehicle detection accuracy from the UAV perspective is the highest. However, the addition of the proposed method slightly decreases the detection performance of the original network in the daytime. It is speculated that there is still much additional infrared information in the daytime scene, which masks the recognizable visible features. Therefore, it is meaningful to design a more flexible adjustment function.

## 5. Conclusions

In this work, a local adaptive illumination-driven input-level fusion for infrared and visible object detection is proposed. The proposed LAIIFusion could significantly reduce the computation and achieve satisfactory FPS in visible–infrared detection. In particular, by dividing the image into multiple grid cells, perceiving the illumination in each grid cell, and redesigning labels with histograms, more accurate scene illumination can be extracted. By using this method, a reliable reference for subsequent fusion could be obtained. Furthermore, an end-to-end offset estimation module is designed for the infrared–visible object location offset problem. This module could efficiently sense the offset through multilayer convolution, effectively alleviating the negative impact of the image on the inter-offset. From the experimental results, it can be seen that the proposed LAIIFusion can generate fused images that are more conducive to detection. Meanwhile, several single-modal object detection networks combined with our method were utilized to conduct performance improvement experiments. The results show that the comprehensive performance is improved, especially in the night detection scene. Through the comparison experiments with the feature-level fused detection networks, it could be concluded that the proposed method only shows less accuracy and obtains faster inference speeds.

At present, the detection performance of input-level fusion methods is still not as high as that of feature-level fusion methods. The complementary information of infrared and visible images has not yet been fully explored. The computational cost of this method is still high for embedded devices. In the next step, we will explore how to effectively extract complementary information from infrared and visible images and further compress existing multimodal networks using a lightweight method.

**Author Contributions:** Conceptualization, J.W. and T.S.; methodology, J.W.; software, Q.W.; validation, Q.W., J.S. and K.Z.; formal analysis, J.W.; investigation, Z.T.; resources, T.S. and Q.W.; data curation, J.W.; writing—original draft preparation, J.W.; writing—review and editing, T.S., Q.W. and J.S.; visualization, J.W. and Z.T.; supervision, Q.W. and J.S.; project administration, T.S. and K.Z.; funding acquisition, T.S. and Q.W. All authors have read and agreed to the published version of the manuscript.

**Funding:** This research was supported in part by the Youth Project of the National Natural Science Foundation of China under Grant 62201237, in part by the National Natural Science Foundation under Grant 61971208, in part by the Yunnan Fundamental Research Projects under Grant 202101BE070001-008, and by the Fundamental Research Project of Yunnan Province (No. 202101AU070050).

**Data Availability Statement:** The DroneVehicle remote sensing dataset was obtained from https://github.com/VisDrone/DroneVehicle, accessed on 29 December 2021. The KAIST pedestrian dataset was obtained from https://github.com/SoonminHwang/rgbt-ped-detection/tree/master/data, accessed on 12 November 2021.

**Conflicts of Interest:** The authors declare no conflict of interest.

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
