# Peer review of "Local Adaptive Illumination-Driven Input-Level Fusion for Infrared and Visible Object Detection"

_remotesensing, doi:10.3390/rs15030660_

Round 1

Reviewer 1 Report

In this work, the authors proposed local adaptive illumination-driven input-level fusion module (LAIIFuison) which effectively improves the around-the-clock detection performance of single modality object detection models and achieve satisfactory speed. The method is interesting, and the results are impressive. However, the manuscript could still be further improved by fixing the following problems:

(1) The author uses two data sets to verify the effectiveness of the proposed method. Why do the two datasets use different training parameters?

(2) It is hoped that more single modality object detection results can be provided in section 3.4.3.

(3) Table 3 distinguishes the fusion stages. But there is no distinction in Table 1. Can the form of Table 1 be similar to Table 3?

(4) Figure 5 provides less detail. Can you give the number of input/output channels and other parameters of Bottleneck module in Figure 5?

(5) There are some detail problems in the manuscript. For example:

1. In the section 3.2, “data enhancement methods” should be expressed as “data augment methods”.

2. In formula (13) and (14), the denominator part does not need bracket.

3. In the section 4.2, please note that the first letter of the word in the title is capitalized.

Reviewer 2 Report

The authors have proposed a visible and infrared object detection method using input-level fusion rather than existing feature-level or decision-level fusion. Precisely, it consists of illumination perception, offset estimation, and adaptive fusion modules. In addition, the authors have conducted multiple experiments to support their claims. However, some concerns remain regarding the proposed method.

1. The input-level fusion has the disadvantage of not appropriately utilizing COCO pre-trained weights, unlike the existing feature-level fusion. How did you solve this problem?

2. The infrared weight W_{inf} is automatically calculated by W_{rgb} in the illumination perception module (W_{inf} = 1-W_{rgb}). It means that if the illumination perception value W_{illum} of the visible image is quite large, the model mainly predicts with the visible image. Even if the illumination of the visible image is large, the influence of the infrared image could be important. It could be a disadvantage of illumination-based input-level fusion.

3. The role of W_{0} is ambiguous. You should explain W_{0} a bit more clearly.

4. On the offset estimation module, it only estimates the shift s_{x} and s_{y}. In practice, IR and visible cameras have different resolutions and FOV, so it is common to consider scale to align the pair images I_{vis}, I_{ir}. It could be another reason why input-level fusion is challenging. 

5. In Table 5, there is no ablation study for the adaptive fusion module, but it seems necessary.

6. In Equation 1, W_{i} comes out as a result of F_{IP}, but there is no specific explanation for this. In Figure 2, the output of the illumination perception module shows only W_{vis} and W_{ir}. In Equation 7, there is no explanation of how to calculate W_{i} in detail.

7. On page 6, line 236 and Figure 4: N seems to be 32. How did you set the value? Are there any experimental results when N is a different value? 

8. Too many typos and errors:

i) On page 1, line 19: UVA -> UAV

ii) On page 2, line 51: [18] reference link is missed.

iii) On page 2, line 51: [18].Liu et al. ~ -> [18]. Liu et al. ~

iv) On page 2, line 70: UVA -> UAV

v) On page 3, line 131: bottomleneck -> bottleneck

vi) On page 4, Figure 2: W_{vis} should be connected to I_{vis}, and W_{ir} should be connected to I_{ir}.

vii) On page 6, line 218: infrared and visible -> visible and infrared

viii) On page 6, line 239: W_{label} is missing on Equation (7).

ix) On page 7, line 250 and Equation (8): X_{vis} or X_{rgb}? Select one of them.

x) On page 4, 5, and 6: W_{vis} or W_{rgb}? W_{ir} or W_{inf}? Select one of them.

xi) On page 15, Table 5: YOLOv5L +IP+OE -> YOLOv5L + LIP + OE

xii) On page 4, Equation 1: I_{ds_{vis}} -> I_{ds_vis}

Round 2

Reviewer 2 Report

Additional experiments have been added to verify the ablation study and hyperparameters of the proposed methods.

The detailed explanation of the proposed methods has been explained clearly.

I am satisfied with the revised manuscript.